# Speed of Light in Hollow-Core Photonic Bandgap Fiber Approaching That in Vacuum

**DOI:** 10.3390/s24216954

**Published:** 2024-10-30

**Authors:** Xiaolu Cao, Mingming Luo, Jianfei Liu, Jie Ma, Yundong Hao, Yange Liu

**Affiliations:** 1School of Electronic and Information Engineering, Hebei University of Technology, Tianjin 300401, China; 202231903062@stu.hebut.edu.cn (X.C.); jfliu@hebut.edu.cn (J.L.); jie.ma@hebut.edu.cn (J.M.); 2Hebei Key Laboratory of Advanced Laser Technology and Equipment, Tianjin 300401, China; 3Institute of Modern Optics, Nankai University, Tianjin 300350, China; hydnku@163.com (Y.H.); ygliu@nankai.edu.cn (Y.L.)

**Keywords:** hollow-core photonic bandgap fiber, Fresnel reflection, optical frequency domain reflectometry, light speed measurement

## Abstract

A Fresnel mirror is introduced at a hollow-core photonic bandgap fiber end by fusion splicing a short single-mode fiber segment, to reflect the light backward to an optical frequency domain reflectometry. The backward Fresnel reflection is used as a probe light to achieve light speed measurement with a high resolution and a high signal-to-noise ratio. Thus, its group velocity is obtained with the round-trip time delay as well as the beat frequency at the reflection peak. Multiple Fresnel peaks are observed from 2180.00 Hz to 13,988.75 Hz, corresponding to fusion-spliced hollow-core fiber segments with different lengths from 0.2595 m to 1.6678 m, respectively. The speed of light in the air guidance is calculated at 2.9753 × 10^8^ m/s, approaching that in vacuum, which is also in good agreement with 2.9672 × 10^8^ m/s given by the numerical analysis with an uncertainty of 10^−3^. Our demonstration promises a key to hollow-core waveguide characterization for future wide-bandwidth and low-latency optical communication.

## 1. Introduction

As a promising solution to a primary concern in mass data transmission, hollow-core microstructured optical fibers (HC-MOFs) are expected to break through the capacity limit of commercial single-mode fibers (SMFs) [1,2,3]. The HC-MOF is considered as an ideal combination of a free-space light transmission and optical waveguide, for its low latency, low dispersion, and high threshold and flexibility [4,5,6], which shows prospects in high-capacity signal and energy transfer [7,8].

Due to the strong confinement over 99.8% in the air core, the light propagates in HC-MOFs at a group velocity approaching that in vacuum [6]. However, achieving hollow-core fiber leakage, scattering, and micro-bending loss levels comparable to or lower than conventional fibers has proved to be a difficult challenge until recently [9,10,11]. Moreover, subject to the distortion in nano-scaled structures during the manufacturing, the intrinsic characteristics of the optical waveguide are not convincing enough just with those results from a numerical analysis. As an example, light speed in HC-MOFs becomes a crucial and tough issue to be addressed, where plenty of smart schemes and impressive techniques were taken [12,13,14,15,16,17,18,19,20].

The direct measurement of group velocity is recognized as time delay evaluation using modulated probe light, which is improved but still limited in accuracy by loss, distances, response time, and dispersion [12,13,14]. For the poor signal-to-noise ratio (SNR) caused by higher loss in HC-MOFs, the performances degenerate in the time-of-flight technique [14], re-circulating loop [15], and optical network tester [16]. From the grating Bragg wavelength, the group velocity can be determined, but there are errors due to the uncertainty in pitch of the phase masks used for grating inscription [17]. In the low-coherent interferometry method, the group velocity can be derived by measuring the group velocity dispersion (GVD) over a continuous spectral range. However, the dynamic range is limited by dispersion with a low power and excessive bandwidth source. Also, the spatial alignment is hard to achieve at a high precision from its complexity and instability [18,19,20]. Due to the high output light intensity, the methods abovementioned result in a poor SNR using forward transit light. Thus, an all-fiber optical frequency domain reflectometry (OFDR) stands out using backward Fresnel reflection as the probe light, whose light intensity is 3-4 orders higher than that of the background Rayleigh scattering [21]. It brings a high-resolution and a high-SNR characterization method for mode identification and evolution in microstructured optical fibers (MOFs) [22,23,24]. Moreover, group birefringence in MOFs can also be measured and regulated using OFDR, where the polarized Fresnel reflection between the silica core and free space is easy to identify at the fiber end face [25,26]. Due to the absence of the refractive index (RI) gradient at the fiber end, however, direct group velocity measurement with OFDR is not applicable in HC-MOFs when compared with solid core fibers.

In this paper, the backward reflection at the man-made Fresnel mirror in a hollow-core photonic bandgap fiber (HC-PBF) is harvested with the OFDR, in which the group velocity can be obtained with the related time delay and the beat frequency. Several hollow-core fiber segments with different lengths from 0.2595 m to 1.6678 m are fusion-spliced to a single-mode fiber end face, leading to their beat-frequency differences from 2180.00 Hz to 13,988.75 Hz, respectively. Thus, considering the distortion between the actual HC-PBF and its ideal model, the averaged group velocity in the HC-PBF is calculated as 2.9753 × 10^8^ m/s, which is close to 2.9672 × 10^8^ m/s given by the numerical analysis. Our demonstration provides a universal and flexible method for light speed measurement in air guidance, which is achieved just within a few meters of a fiber segment and applicable to all kinds of HC-MOFs as well.

## 2. Principle and Experimental Setup

The spliced SMF-HCPBF joint is introduced as a Fresnel mirror for group velocity measurement as shown in Figure 1a. Notably, the Fresnel mirror is deliberately manufactured at the HC-PBF end to reflect the light backward to the OFDR, for the absence of the RI gradient between the air core and free space. To avoid the excitation of the surface super-modes in HC-PBF, the pig-tail SMF1 in OFDR is mechanically aligned with its 3D adjustment in the fusion splicer.

Figure 1b shows a schematic diagram of our experimental setup for group velocity measurement in HC-PBF based on OFDR [22]. Frequency-swept light from the tunable laser source (TLS) is split by a 20/80 polarization-maintaining coupler (PMC), PMC1. The 20% optical power goes into an auxiliary interferometer consisting of a 1:1 coupler (CP) and two Faraday Rotation Mirrors (FRMs), which generates a sine reference collected by a balanced photo detector (BPD), BPD1, for sweeping nonlinearity calibration. In the main interferometer, the remaining 80% is split once again by another 20/80 PMC2. Most of the optical power enters the fiber under testing (FUT) from port 1 to port 2 through a circulator. Meanwhile, the rest enters a reference path with a 2 m delay fiber to make up the optical path difference between the reference and test paths. Both the backward reflection from port 3 and the local light from the reference path split into S/P polarizations by two polarization beam splitters (PBSs), respectively. The polarized beams interfere at PMC3 and PMC4 according to their polarizations, and the interferences are then collected by BPD2 and BPD3 separately. The bandpass signals from the three BPDs are acquired by a 4-channel digital data acquisition (DAQ), for post processing once the frequency sweep begins.

Figure 2 shows an extremely small group dispersion at 10^−6^ of the fundamental mode (LP_01_ mode) in HC-PBF, with its ideal and actual transverse cross-sections shown as the insets. A 7-cell hollow core is surrounded by 8 rings of cladding holes to form the multilayer reflection microstructures and photonic band gaps. The phase refractive index is cosine projection along the axial direction and lower than that in vacuum, effectively confining the light transmission in the air core. In the numerical analysis, the dimensions of the cross-section model are adjusted according to the microscopic image, where the pitch (*Λ*, distance between two adjacent holes) is 4.1 μm, the core radius (*R*) is 7.9 μm, and the cladding air hole diameter-to-pitch ratio (*d*/*Λ*) is 0.968. The hexagonal air hole circular angle diameter-to-pitch ratio (*dc*/*Λ*) is 0.3, the core filet diameter-to-pitch ratio (*Dc*/*Λ*) is 0.5, and *t* refers to the wall thickness of the central air core. The material RI of the fusion silica background is expressed by the Sellmeier equation [27]. Thus, the wavelength-dependent phase RI curve is depicted with data from Finite Element Method (FEM) COMSOL Multiphysics, and the group RI of the LP_01_ mode is further calculated with Equation (1).
(1)$$n_{g}(\lambda )=n_{eff}(\lambda )-\lambda \cdot dn_{eff}(\lambda )/d\lambda ,$$
where *n_g_* and *n_eff_* represent the group RI and phase effective RI, respectively, and *λ* is the wavelength. Subsequently, the group velocity *v_g_* and the time delay *τ* can be calculated with Equation (2).
(2)$$v_{g}=\frac{L}{\tau }=c{\left(\frac{d\beta }{dk}\right)}^{-1}=\frac{c}{n_{g}}=\frac{c}{n_{eff}-\lambda \cdot dn_{eff}/d\lambda },$$
where *L* is the fiber length, *c* is the speed of light in vacuum, *β* is the propagation constant, and *k* is the wavevector in vacuum. Accordingly, the group velocity of the LP_01_ mode is theoretically obtained at 2.9672 × 10^8^ m/s with the time delay 3.3702 μs/km. Considering the expansion and distortion of the hollow core and innermost air holes, the actual light speed in the air guidance might be faster than that in simulation. Thus, the group velocity increases 1.6862‰ of the air guidance when the diameter of the hollow core expands by 1%.

When frequency-swept light reaches the HCPBF-SMF2 joint, significant reflection occurs at the Fresnel mirror due to the large RI gradient. Thus, the group velocity *v_g_* in hollow-core fibers is obtained with the time delay *τ* and the beat frequency *f_Beat_* at the Fresnel reflection peak with Equation (3).
(3)$$v_{g}=2L/\tau =2L\gamma /f_{Beat},$$
where 2*L* refers to the round-trip propagation in HC-PBF, and *γ* is the scanning rate of the laser.

For an actual OFDR system, however, the total time delay includes delay *τ_sys_* in the OFDR system, *τ*_*S*1_ in the SMF “pig-tail” of length *L_S_*_1_, *τ_HC_* in HC-PBF of length *L_HC_*, and *τ_S_*_2_ in Fresnel mirror SMF of length *L_S_*_2_. Thus, the total beat frequency related to time delay can be expressed with Equation (4).
(4)$$\begin{array}{ll} f_{Beat} & =f_{Beat}^{sys}+f_{Beat}^{S1}+f_{Beat}^{HC}+f_{Beat}^{S2}\\ & =\gamma (\tau_{sys}+\tau_{S1}+\tau_{HC}+\tau_{S2})\\ & =\gamma (\tau_{sys}+2L_{S1}/v_{g}+2L_{HC}/v_{g}^{LP_{mn}}+2L_{S2}/v_{g}). \end{array}$$

Figure 3 shows the beat-frequency differences from 2184 Hz to 14,037 Hz corresponding to HC-PBFs with different lengths applied in the experiment from 0.2595 m to 1.6678 m. By numerical simulation based on the OFDR principle, the beat-frequency difference also shows a linear relation to the fiber length, providing a method for group velocity measurement in air-guidance hollow-core fibers with Equation (3).

## 3. Results and Discussion

The Santec tunable semiconductor laser TSL-570 is set to 10 mW at a scanning rate of 10 nm/s (*γ* = 1.2487 THz/s), within a spectral range from 1540 nm to 1560 nm. As the Fresnel mirror mentioned above in Figure 1b, a short SMF segment is fusion-spliced to the HC-PBF with different lengths. The other side of the HC-PBF is mechanically aligned with the SMF pig-tail using a fusion splicer, Fujikura 80s, Japan, to ensure the reflection back to the OFDR. The multi-component interferences are expanded in the frequency domain by the Fast Fourier Transform (FFT) method, where the reflections at different positions are clearly distinguished in Figure 4a. The external port of the OFDR is located at 1483.13 Hz, and the next on the right is the mechanical alignment between the SMF1 and HC-PBF at 16,218.13 Hz. The Fresnel mirror presents itself at the HCPBF-SMF2 joint of 24,626.25 Hz, where the end face of the SMF2 shows at 26,118.13 Hz. By local zooming at the HCPBF-SMF2 joint in Figure 4b, the Fresnel peak of the LP_01_ mode is observed with remaining energy of the surface super-modes or high-order modes on the right.

Several HC-PBF segments with different lengths are further tested in the same way. The beat-frequency difference is calculated with the positionings *N*_1_ at the mechanical alignment and *N*_2_ at the Fresnel mirror, from the beginning to end of the HC-PBF expressed by Equation (5).
(5)$$f_{Beat}^{HC}=(N_{2}-N_{1})\Delta f=(N_{2}-N_{1})f_{s}/N,$$
where Δ*f* represents the OFDR frequency resolution of 0.62 Hz, and *f_s_* and *N* are the sampling rate of 2 M Hz and sampling point of 3.2 M separately set by DAQ.

The beat-frequency differences show a linear relation to their lengths of the HC-PBFs, as the black scatters with error bars shown in Figure 5. The measured group velocity and time delay of the LP_01_ mode are averaged at 2.9753 × 10^8^ m/s and 3.3610 μs/km in HC-PBF separately, which are in good agreement with those in the numerical analysis. Subject to deformation during the manufacturing and the deviation in fiber length measurement, the deviation in $v_{g}^{{\mathrm{LP}}_{\mathrm{01}}}$ becomes larger along the increasing fiber length shown as the blue error bars and listed in the inset table of Figure 5. Thus, an uncertainty of the light speed in HC-PBF is 10^−3^ of 2.9753 × 10^8^ m/s with a good linearity.

## 4. Conclusions

To measure the speed of light guided in an air core approaching that in vacuum, a Fresnel mirror is introduced at the hollow-core fiber end to reflect the light back to the OFDR. Multiple Fresnel reflection peaks are observed at different beat frequencies from 2180.00 Hz to 13,988.75 Hz, corresponding to different round-trip time delays over fiber segments with different lengths from 0.2595 m to 1.6678 m. Finally, considering the deformation of the cross-section and deviation in length measurement, the group velocity is calculated as 2.9753 × 10^8^ m/s approaching that in vacuum. Our demonstration promises an accurate and flexible method for group velocity measurement, which is beneficial to understand the waveguide characteristics of specific optical fibers. This OFDR-based instrument is easy to operate and applicable just within a few meters of hollow-core fiber for future mass data optical communication.

## Figures and Tables

**Figure 1 sensors-24-06954-f001:**
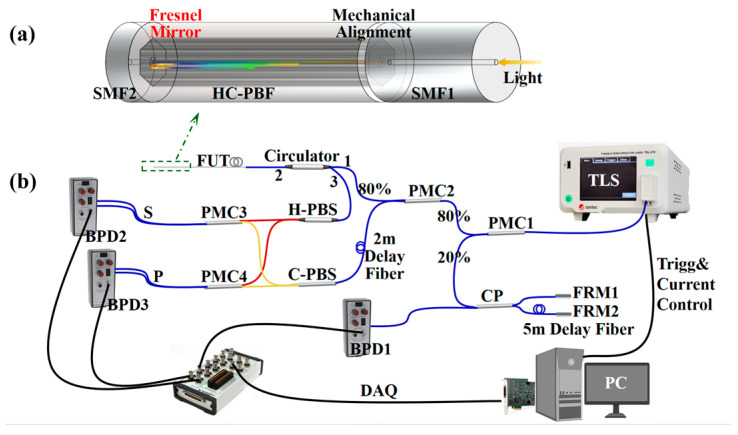
(**a**) A perspective view of a HC-PBF sample with a Fresnel mirror. (**b**) The schematic setup of the dual-polarization coherent OFDR system.

**Figure 2 sensors-24-06954-f002:**
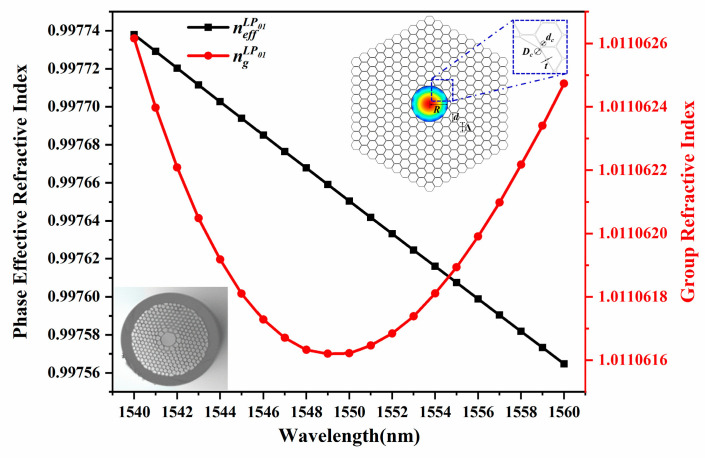
The phase and group dispersion curves of the LP_01_ mode in HC-PBF with the microscopic image, model of the cross-section, and LP_01_ mode profile as insets.

**Figure 3 sensors-24-06954-f003:**
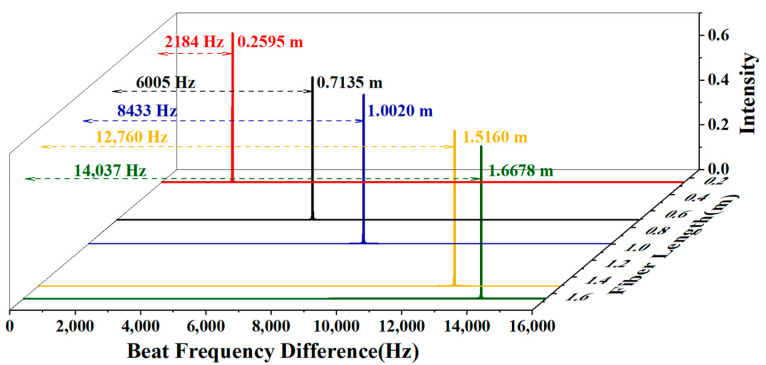
The numerical simulation of the beat-frequency difference using HC-PBF segments with different lengths based on the OFDR principle.

**Figure 4 sensors-24-06954-f004:**
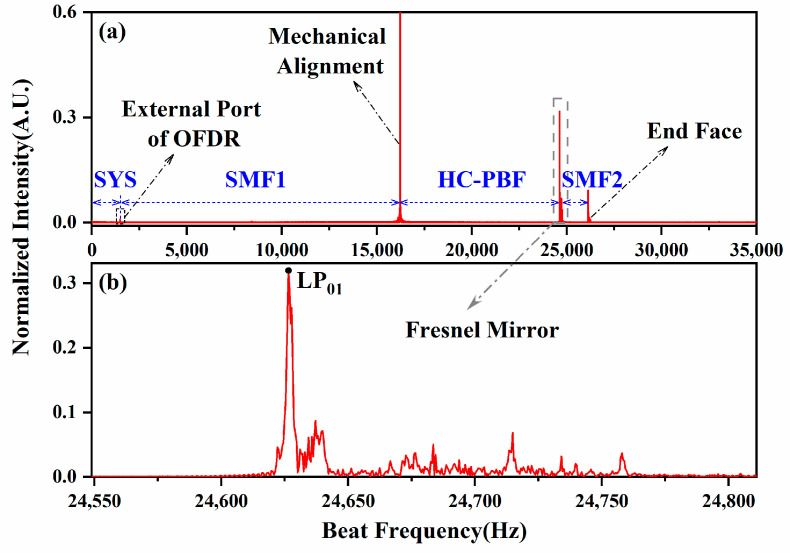
The observation of the Fresnel reflection peak in the frequency domain with 1.0020 m long HC-PBF as an example in (**a**) the whole frequency range and (**b**) the detailed frequency range.

**Figure 5 sensors-24-06954-f005:**
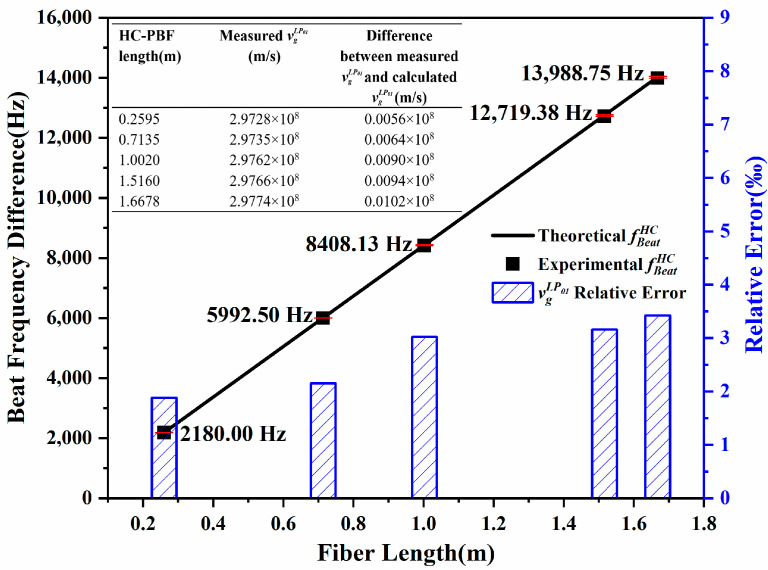
The uncertainty analysis between the numerical analysis and experimental results.

## Data Availability

Data underlying the results presented in this paper are not publicly available at this time but may be obtained from the authors upon reasonable request.

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
