# Peer review of "Speed of Light in Hollow-Core Photonic Bandgap Fiber Approaching That in Vacuum"

_sensors, 2024, doi:10.3390/s24216954_

Round 1

Reviewer 1 Report

Comments and Suggestions for Authors

   This paper presents an interesting method for measuring the speed of light in hollow-core photonic bandgap fibers (HC-PBF) with high accuracy and resolution. The authors successfully achieve precise group velocity measurements by introducing a Fresnel mirror at the end of the HC-PBF and utilizing optical frequency domain reflectometry (OFDR).  After reading the manuscript, the reviewer thinks some major issues in this manuscript need to be addressed.

 1. The numerical analysis has been carried out in a standard FEM solver such as COMSOL? Or was it an in-house developed model? This should be clearly stated in the paper.

2. How was the SMF spliced on the HCF?  Was the SMF just cleaved or was it flat polished? This is the key element of the proposed measurement approach and therefore I would expect more information on the HCF-SMF preparation.

3. References should be described in the Microsoft Word template for Sensors.

Comments on the Quality of English Language

There are some grammatical and format mistakes. Please check the manuscript carefully.

Reviewer 2 Report

Comments and Suggestions for Authors

I am not sure about the benefit of this paper. A new method of measuring the speed of light? Please explain the advantages.

Reviewer 3 Report

Comments and Suggestions for Authors

The manuscript presents a promising demonstration of measuring the group velocity of light in hollow-core photonic crystal fiber using optical frequency domain reflectometry (OFDR). The authors have developed an efficient group index measurement technique for a short length (below 2 m) of hollow-core fiber, paving the way for a widely acceptable measurement technique in optical communications applications. While the study may lack significant novelty in optical experiments and sensing capabilities, the application of OFDR to hollow-core fiber has its practical value. I have additional comments below:

I suggest some of these references for previous works on group index measurement:

  1. B. L. Danielson, "Precise length measurements in multimode optical fibers," Appl. Opt. 30, 3867-3872 (1991).

  2. T.-J. Ahn and D.Y. Kim, "High-resolution differential mode delay measurement for a multimode optical fiber using a modified optical frequency domain reflectometer," Opt. Express 13, 8256-8262 (2005).

  3. GROUP VELOCITY DISPERSION MEASUREMENT USING SUPERCONTINUUM PICOSECOND PULSES GENERATED IN AN OPTICAL FIBRE, K. Mori, T. Morioka and M. Saruwatari.

  4. J. L. Cruz, Y. O. Barmenkov, A. Díez, and M. V. Andres, "Measurement of phase and group refractive indices and dispersion of thermo-optic and strain-optic coefficients of optical fibers using weak fiber Bragg gratings," Appl. Opt. 60, 2824-2832 (2021).

1. The accuracy of the measured group index or velocity is not great.

2. As length increases, the accuracy of results decreases (Fig. 5).

3. The complete form of FFT is missing in the manuscript.

4. I suggest adding a reason for the phase refractive index below 1 or the vacuum refractive index(fig. 2) for a broad readership. 

Overall, the manuscript's flow is good, and it’s well-presented and articulated. The results appear credible. The technique presented is old, but applying it to hollow-core photonic crystal fiber seems novel. If the authors find or point to significant novelty in this work by comparing it with previous works in more detail, this manuscript can be recommended for publication.

Round 2

Reviewer 2 Report

Comments and Suggestions for Authors

This paper provides speed of light measurement in a hollow core optical fiber. This is a useful detail for using this fiber. 

I have to ask: why are only 4 references out of 27 that are not lined through remaining? Although these 4 are relevant to the work, why are the other 23 lined through? This fact reduces my confidence in the work.

Author Response

During the initial review round, one of the reviewers gave the opinion of "References should be described in the Microsoft Word template for Sensors". Consequently, we adjusted the original 23 references to align with the formatting requirements specified in the Microsoft Word template for Sensors and presented them with red lines. Additionally, another reviewer gave the opinion of "I suggest some of these references for previous works on group index measurement" in the initial review round, so we added another 4 references related to the work as required by the reviewer. As these are supplementary references, they are not lined but highlighted in red. Furthermore, since these 4 references have been placed in their respective appropriate positions within the manuscript, the reference labels are sequentially adjusted.

Reviewer 3 Report

Comments and Suggestions for Authors

Although the accuracy in measurement of group index is up to 10^-3 and the accuracy of results decreases with increases in length of hollow core fiber, which are limits posed by the method. However, the demonstrated results are significant enough for a practical value. The article presents details of an optical experiment with hollow core fiber, which has not been performed in the same manner before. The manuscript is acceptable for publication in its form.

Additionally, the authors have answered my remarks in the response letter and revised the manuscript. I am satisfied with the revised manuscript and recommend it for publication in the MDPI Sensors Journal.

Author Response

We thank the reviewer for the positive and constructive comments on our manuscript.